# Surveillance Value of Apparent Diffusion Coefficient Maps: Multiparametric MRI in Active Surveillance of Prostate Cancer

**DOI:** 10.3390/cancers15041128

**Published:** 2023-02-10

**Authors:** Aleksandar Georgiev, Lyubomir Chervenkov, Mladen Doykov, Katya Doykova, Petar Uchikov, Silvia Tsvetkova

**Affiliations:** 1Department of Diagnostic Imaging, Medical Faculty, Medical University Plovdiv, Bul. Vasil Aprilov 15A, 4002 Plovdiv, Bulgaria; 2Department of Diagnostic Imaging, Complex Oncology Center Plovdiv, ul. Pere Toshev 62, 4004 Plovdiv, Bulgaria; 3Research Complex for Translational Neuroscience, Medical University of Plovdiv, Bul. Vasil Aprilov 15A, 4002 Plovdiv, Bulgaria; 4Department of Urology and General Medicine, Medical Faculty, Medical University Plovdiv, Bul. Vasil Aprilov 15A, 4002 Plovdiv, Bulgaria; 5Department of Special Surgery, Medical Faculty, Medical University Plovdiv, Bul. Vasil Aprilov 15A, 4002 Plovdiv, Bulgaria

**Keywords:** prostate cancer, screening, imaging, surveillance, mp-MRI

## Abstract

**Simple Summary:**

Prostate cancer is one of the leading oncological diseases in the male population. Early detection and treatment of the disease gives better survival chances and a better long-term prognosis. The aim of this study is to establish the value of magnetic resonance imaging sequences for the detection and active surveillance of prostate cancer. The collected data from 530 patients indicate that the apparent diffusion coefficient (ADC) is the most specific and sensitive magnetic resonance sequence. Therefore, we advocate for its inclusion in the routine scanning protocol for prostate cancer. We advise on at least a bi-parametric magnetic resonance that includes apparent diffusion coefficient and a T2 sequence.

**Abstract:**

Background: This study aims to establish the value of apparent diffusion coefficient maps and other magnetic resonance sequences for active surveillance of prostate cancer. The study included 530 men with an average age of 66, who were under surveillance for prostate cancer. We have used multiparametric magnetic resonance imaging with subsequent transperineal biopsy (TPB) to verify the imaging findings. Results: We have observed a level of agreement of 67.30% between the apparent diffusion coefficient (ADC) maps, other magnetic resonance sequences, and the biopsy results. The sensitivity of the apparent diffusion coefficient is 97.14%, and the specificity is 37.50%. According to our data, apparent diffusion coefficient is the most accurate sequence, followed by diffusion imaging in prostate cancer detection. Conclusions: Based on our findings we advocate that the apparent diffusion coefficient should be included as an essential part of magnetic resonance scanning protocols for prostate cancer in at least bi-parametric settings. The best option will be apparent diffusion coefficient combined with diffusion imaging and T2 sequences. Further large-scale prospective controlled studies are required to define the precise role of multiparametric and bi-parametric magnetic resonance in the active surveillance of prostate cancer.

## 1. Introduction

Active surveillance requires a management strategy for patients with prostate cancer [1]. Alternative management following a diagnosis of prostate cancer includes radiotherapy (external or brachytherapy) and ablative therapies of the tumor foci, including cryotherapy, high-intensity focused ultrasound (HIFU), and irreversible electroporation (IRE), as well as surgical removal of the whole prostate and seminal vesicles (radical prostatectomy). The benefits of multiparametric magnetic resonance imaging (mp-MRI) for active surveillance are that it reduces the need for repeat digital rectal examinations and biopsies, which cause discomfort and increase the risk of infections. The lack of invasiveness of the method leads to a good acceptance rate amongst patients. The Prostate Imaging-Reporting and Data System (PI-RADS) is aimed at standardizing the visual description of prostate cancer characteristics. Mp-MRI is routinely used by clinicians before biopsy and for the pre-operative stage of the disease [2]. Mp-MRI is becoming the imagined gold standard for suspicious (PI-RADS 3) prostate cancer lesions and is being increasingly accepted by practitioners [3]. Multiparametric MRI enables prostate cancer detection, localization, and further characterization in terms of tumor size and stage [4,5,6,7,8,9]. Multiparametric MRI may be helpful in two stages of the active surveillance protocol, as the baseline examination at patient enrolment [5,6,7,8,9] and an alternative to follow-up prostate biopsy during the active surveillance program. The collection of candidates for active surveillance and disease reclassification during active surveillance using imaging of the prostate gland would be beneficial, increase diagnostic accuracy, and improve patient management. Therefore, there is a need for additional clinical studies regarding the role of mp-MRI in active surveillance programs.

The aim of this study is to establish the value of apparent diffusion coefficient (ADC) maps and other magnetic resonance imaging (MRI) sequences for active surveillance of prostate cancer and the ability to detect clinically significant cancer.

## 2. Materials and Methods

The present study included 530 patients, aged 44 to 82 years, who were examined for prostatic cancer (PC) at the Urology department of the University General Hospital “Kaspela” in Plovdiv, Bulgaria, and the Complex Oncology Center Plovdiv, Plovdiv, Bulgaria in the period from March 2021 to August 2022. The study was conducted in accordance with the Declaration of Helsinki and approved by the Committee for Scientific Ethics at the Medical University of Plovdiv.

The inclusion criteria were: (1) elevated prostate-specific antigen (PSA) levels and/or suspicious tumor evidence at a clinical examination; (2) signed informed consent forms for participation in the study. The following exclusion criteria were applied: (1) refusal to give informed consent; (2) refusal for biopsy; (3) refusal for mp-MRI examination; (4) terminal patients or patients in critical condition.

Prior to mp-MRI and biopsy, all patients underwent digital rectal examination (DRE) and transrectal ultrasonography. Multiparametric MRI exams were performed on 3T MRI. Pathological changes were classified according to the Prostate Imaging-Reporting and Data System (PI-RADS). Each lesion was classified from 1 to 5 according to the possibility of cancer. The five scores are: PI-RADS 1—Very low possibility of cancer. PI-RADS 2—Low possibility of cancer. PI-RADS 3—Intermediate possibility of cancer (suspicious cancer lesion). PI-RADS 4—High possibility of cancer. PI-RADS 5—Very high possibility of cancer.

Transperineal biopsy (TPB) was performed in all cases with sedation and cognitive (visually estimated) fusion technique. About 20–22 cores are taken from 16 to 18 anatomical locations of the prostate gland. All patients were given 2 tablets Ciprofloxacin (500 mg) along with one tablet Tamsulosin (0.4 mg) one night before the biopsy. Post-biopsy, the two oral medicaments were continued for four days to reduce infection rates and the possibility of abscess formation.

The obtained biopsy material was classified according to the Gleason scale. This scale classifies tumors according to their histological appearance—from 1 to 5, with grade 5 referring to the most abnormal-looking tissue. Most tumors express two histological types, and the total score is expressed as the sum of these two most common types. The highest possible Gleason score is 10, and a score below 6 indicates a benign lesion.

The statistical analysis was performed using SPSS version 27.0 (SPSS Inc., Chicago, IL, USA). The continuously measured variables were screened for normality through the Shapiro-Wilk test and were described with the median values and the interquartile ranges (IQR). The categorical variables were presented as frequencies and percentages (%). The receiver operating characteristic (ROC) curve was used to assess the level of agreement between mp-MRI and biopsy in active surveillance of prostate cancer. All tests were two-tailed, and the results were interpreted as significant at Type I error alpha = 0.05 (*p* < 0.05).

## 3. Results

The study included 530 men with an average age of 66, ranging between 44 to 82 years. The data from PSA testing, multi-parametric magnetic resonance imaging, and transperineal biopsy are given in Table 1. The PSA values ranged from 2 ng/mL to 183 ng/mL, with a median of 8.40 ng/mL. Cancer-appearing lesions were found in 430 (81.10%) of the patients and were most frequently located in the peripheral zone and in the middle part of the prostate gland. All 430 lesions were positive on T2 weighted images, of which 270 (62.80%) were in the peripheral zone (PZ) and 160 (37.20%) in the transition zone (TZ). Positive diffusion-weighted imaging (DWI) sequences found 400/430 lesions, of which 260 (65.00%) were in PZ and 140 in TZ (35.00%). Thirty lesions were positive on dynamic contrast-enhanced (DCE) sequences, 200 (66.70%) in PZ and 100 (33.30%) in TZ. 390 lesions were found positive on ADC, of which 250 (64.10%) were in PZ and 140 (35.90%) in TZ. According to the Prostate Imaging–Reporting and Data System (PI-RADS v.2), 340 (61.10%) patients were categorized as likely or very likely to have clinically significant cancer. TBP showed positive results for 350 out of the 430 patients with suspicious lesions (66.00%).

Of the 90 suspicious cases (PI-RADS level 3), 50 (55.60%) had a Gleason score of 6 or less, and 40 (44.40%) had a Gleason score of 7. Of the 210 cases classified in PI-RADS level 4, 200 (95.20%) had a Gleason score of 7 or above, and 10 (4.80%) had a Gleason score below 6. Of the 130 cases that were classified as very likely to have clinically significant cancer (PI-RADS level 5), 120 (92.30%) had a Gleason score of 7 or above and 10 (7.70%) had a Gleason score of 6 or below. As a whole, 20 out of 340 (5.90%) patients classified in PI-RADS levels 4 and 5 showed Gleason score results of 6 or lower.

We observed a level of agreement of 67.30% concordance between the ADC maps, PI-RADS classification of the lesions, and the TPB, with a sensitivity of 97.14%, a specificity of 37.50%. According to our data ADC is the most accurate sequence, followed by DWI in PC detection (Table 2).

## 4. Discussion

### 4.1. Multiparametric MRI Active Surveillance of Prostate Cancer in the Context of Biopsy

Low-risk prostate carcinoma can be confirmed based on the histological assessment of the prostatectomy specimen, often using sizable whole-mount tissue sections [10]. The preoperative assessment, based on digital rectal examination, measurement of PSA level, and repeat prostate biopsy have significant limitations. Reclassification is required in many cancers within two years in up to 20–30% of patients [10,11]. Most of the occurrences of reclassification of prostate tumors are caused by under-sampling at the first biopsy, rather than the progression of an indolent tumor [12]. This indicates that the results of routine serial biopsies may be misleading. Therefore, there is an existing need to further develop noninvasive methods, such as mp-MRI, to reduce the risk of underestimation. One of the main disadvantages of serial biopsies is anterior prostate cancer underdiagnosis, which is a cause of reclassification of 25% of patients undergoing active surveillance after two years [13,14]. The 95% negative predictive value of multiparametric prostate MRI highlights the opportunity to avoid repeat biopsies for active surveillance and monitoring. Studies have shown that MRI-guided prostate biopsy detects 52% of tumors in patients with prior negative serial biopsies [15,16]. MR-transrectal ultrasound (MR-TRUS) image fusion targeted biopsy can identify more cancers per core than serial transrectal ultrasound (TRUS)-guided biopsy, regardless of the location of the lesion (anterior or posterior). The inadequate sampling of prostate tumors on a serial biopsy is a reason for incorrect grading in between 23% and 25% of cases [16,17].

The baseline multiparametric MRI may reduce the number of men under active surveillance and who have an overdiagnosis of insignificant cancers [18]. The accuracy of the detection of prostate cancer using a strategy of biopsy of significant lesions in men with increased PSA levels is more accurate than for serial biopsy (*p* < 0.001) [19,20]. Targeted prostate biopsy identifies 16% more Gleason grade 7 prostate cancers. Based on the Gleason score, multiparametric MRI provides data on tumor volume, location, and grade or behavior. A high negative predictive value of multiparametric MRI may reduce re-biopsy indications. According to our data there is 67.30% concordance between the ADC maps, PI-RADS classification of the lesions, and the TPB, with a sensitivity of 97.14% and a specificity of 37.50%. Therefore, there is a use for multiparametric MRI before inclusion in an active surveillance program, followed by a targeted prostate biopsy to reduce the detection of insignificant prostate cancer and reduce the number of men undergoing active surveillance. The percentage of the patients enrolled in active surveillance protocols, based on a negative prior 12-core TRUS biopsy, who had cancer of the anterior portion of the gland indicated as a suspicious lesion on MRI performed before biopsy and diagnosed with a targeted biopsy was high, at up to 89% [14]. These findings support the value of baseline multiparametric MRI for active surveillance of prostate cancer.

### 4.2. The Value of ADC, DWI, and Other MRI Sequences in Active Surveillance of Prostate Cancer

Diffusion-weighted imaging (DWI) has been the main magnetic resonance sequence for tumor imaging in the peripheral zone of the prostate gland, and T2-weighted imaging is the main sequence for tumors in the transitional zone [8]. Water molecules, the primary source of protons in the body, move freely in normal tissue. Tumors, including prostate cancer, consist of densely packed cells and an abundance of cell membranes with restricted diffusion of water [20,21]. DWI imaging has resulted in the ability to distinguish between low-grade, intermediate-grade, and high-grade prostate cancer [22]. The combined use of MRI findings, clinical data, and biopsy results in active surveillance enrolment has been proposed as early as 2010 [11]. According to previous studies, MRI-based models showed better results than clinical models (*p* < 0.05) [11,23,24]. Previous studies have shown that a decreased ADC value indicates aggressive transformation compared with previous MRI findings [10,11,12,13,14,15,16,17,18,19]. Studies have identified a correlation between the apparent diffusion coefficient values and the histological Gleason grade [25,26]. Our data confirms the value of ADC maps, and the fact that it is the most specific and sensitive MRI sequence for prostate cancer for a Gleason grade 7 or above. Lower ADC values and a higher signal on DWI correspond with the denser structure of more minor differentiated cancers [27,28]. This inverse relationship between the Gleason score and the ADC value was found for prostate cancers in the peripheral zone. In a single parameter setting according to our data ADC, DWI, and T2 scans demonstrate the highest levels of sensitivity. DCE sequences are less informative. The specificity of a single MRI parameter, on the other hand, is still problematic. ADC maps are the most sensitive and specific sequences, while T1 sequences have no practical value as a single MRI parameter. Compared to the combined mp-MRI (75.00%) specificity, we do not advise using single MRI sequence techniques for early detection of prostatic carcinoma. However, we advocate for the inclusion of ADC maps in the routine mp-MRI scanning protocol. Additionally, the baseline ADC value is known to act as an independent predictor for unfavorable findings of control biopsy and time to radical prostatectomy. [27,28].

Mp-MRI is a valuable tool for enrolling patients in active surveillance programs and a monitoring method for patients already under active surveillance. Patients under active surveillance and suspicion of malignancy (PI-RADS 3 or higher) on multiparametric MRI have an increased risk of upgrading the Gleason score compared with patients who did not have imaging findings [25,26]. For lesions deemed high risk for significant prostate cancer (PI-RADS 4 and above), the positive predictive value was between 0.87 and 0.98, which means that, in these cases, repeat biopsy was strongly recommended [28].

### 4.3. Perspective and Future Studies

Recently, diverse machine-learning applications focusing on prostate cancer imaging have been described [28,29]. The proposed solutions based on various algorithms may aid the main problems regarding segmentation of the prostate gland, assessment of lesion aggressiveness to distinguish between indolent and clinically significant cancers, enrollment into active surveillance, detection and diagnosis, and identification of tumor invasion. Future developments of machine-learning algorithms may also be used to identify transition zone and peripheral zone tumors, PI-RADS, reproducibility of image interpretation, the differentiation between malignant and benign conditions, such as benign prostatic hyperplasia, and inflammatory diseases such as prostatitis, as well as local tumor staging.

## 5. Conclusions

Patients who are considered candidates for active surveillance for prostate cancer would benefit from multiparametric MRI, as it enables better initial diagnosis and reduces the need for repeat biopsies. Our data suggest that MRI and PI-RADS value of 4 and above correspond to biopsy findings of neoplasms with a Gleason score of 7 or higher—clinically significant prostate cancer. Multiparametric MRI is a valuable tool for monitoring patients already under active surveillance, or as an initial imaging—like a baseline mammography. Based on our findings, we advocate that an ADC sequence should be included as an essential part of MRI scanning protocols for prostate cancer in at least bi-parametric MRI. The best option will be the apparent diffusion coefficient combined with diffusion imaging and T2 sequences. Further large-scale prospective controlled studies are required to define the precise role of multiparametric MRI and bi-parametric MRI sequences in active surveillance for prostate cancer.

## Figures and Tables

**Table 1 cancers-15-01128-t001:** Clinical characteristics of the patients.

Variables	Statistics
Age (years) ∘ Median (IQR) ∘ Min.—Max.	66 (10) 44–82
PSA ng/mL ∘ Median (IQR) ∘ Min.-Max.	8.40 (5.6) 2.00–183.00
Zone n (%) ∘ Transition (TZ) ∘ Peripheral (PZ) ∘ No lesions	160 (30.20%) 270 (50.90%) 100 (18.90%)
Part n (%) ∘ Apical ∘ Basal ∘ Middle ∘ No lesions	160 (30.20%) 50 (9.40%) 220 (41.50%) 100 (18.90%)
T2 n (%) ∘ Positive ∘ Negative	430 (100.00%) 0 (0.00%)
DWI n (%) ∘ Positive ∘ Negative	400 (93.00%) 30 (6.00%)
DCE n (%) ∘ Positive ∘ Negative	300 (69.70%) 130 (30.30%)
ADC n (%) ∘ Positive ∘ Negative	390 (90.60%) 40 (9.40%)
PI-RADS v. 2 n (%) ∘ Level 1 (Clinically significant cancer highly unlikely) ∘ Level 2 (Clinically significant cancer unlikely) ∘ Level 3 (Clinically significant cancer is equivocal) ∘ Level 4 (Clinically significant cancer is likely) ∘ Level 5 (Clinically significant cancer is very likely)	10 (2.10%) 90 (16.90%) 90 (16.90%) 210 (39.60%) 130 (24.50%)
Transperineal Prostate Biopsy ∘ Positive ∘ Negative Gleason score ∘ Gleason 6 or lower ∘ Gleason 7 ∘ Gleason 4 + 3 = 7 and above	350 (81.40%) 80 (18.60%) 80 (18.60%) 250 (58.10%) 100 (23.30%)

**Table 2 cancers-15-01128-t002:** Receiver operating characteristic (ROC) curve results for the level of agreement between mp-MRI and biopsy in active surveillance of prostate cancer.

mp-MRI vs. Biopsy	AUC	SE	95% CI	*p*	Sensitivity	Specificity
ADC	0.673	0.093	0.492 to 0.855	0.06	97.14%	37.50%
DWI	0.611	0.083	0.448 to 0.774	0.18	97.14%	25.00%
DCE	0.545	0.099	0.350 to 0.739	0.65	71.43%	37.50%
T2	0.500	0.116	0.276 to 0.724	1.000	100.00%	00.00%
PI-RADS	0.832	0.085	0.665 to 0.999	<0.001	91.43%	75.00%

## Data Availability

The data presented in this study are available on request from the corresponding author. The data are not publicly available due to ethical concerns.

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
