# Peer review of "Surveillance Value of Apparent Diffusion Coefficient Maps: Multiparametric MRI in Active Surveillance of Prostate Cancer"

_cancers, 2023, doi:10.3390/cancers15041128_

Round 1
Reviewer 1 Report
I read the manuscript with interest. I have following suggestions -
Active surveillance (AS) is a more accepted term, I suggest replacing active prophylaxis with active surveillance throughout
I disagree that active surveillance protocols don’t use MRI. MRI is now increasingly being used for initial diagnosis and follow up of patients undergoing AS. You can easily find recent references on use of MRI in AS.
Please define all short forms prior to initial mention. I am assuming TPB means trans perineal biopsy
What is the definition of cancer suspicious lesions, pirads 3 or 4.
It is not Rock curve – it is called ROC curve (table 2)
There is no mention of gleason grade throughout the manuscript
It is not clear if purpose was any cancer detection or clinically significant cancer detection (which is Gleason 7 or higher)
The manuscript completely goes directionless and against the intended purpose in introduction. It gives no link as to how MRI helped in AS and rule in or out higher or lower grade gleason disease.
It seems as if manuscript simply proves how MRI is good in cancer diagnosis with no mention about significant or insignificant cancer diagnosis
Author Response
We thank the reviewer for the time spent on the manuscript. We are grateful for the criticism of the paper. That has allowed for a much more focused article and a better-looking publication overall. We hope the revised manuscript has achieved sufficient quality and focus and can be accepted for publication.
We have made all the suggested changes is the manuscript as follows:
- A professional medical editor edited the manuscript.
- An experienced translator checked the English language of the paper.
- The design of the research was improved.
- All methods' description was enhanced.
- The results have been clarified.
- The conclusions have been better supported.
- The term prophylaxis was replaced throughout the paper with surveillance.
- We have added more recent references for MRI in AS. We have made the according changes in the text as well.
- All short forms have been defined before their initial mention in the text.
- Cancer suspicious lesion was defined as Pi-Rads 3. Appropriate changes have been made in the text.
- ROC curve misspelling was addressed.
- Gleason grade values have been added in the Results section. Gleason's score is described in the paper as well.
- The purpose is to identify clinically significant cancer and establish the value of MRI for the detection and surveillance of prostate cancer. We think we have clarified this in the revised version of the paper.
- We have managed to shorten the discussion and focus on the MRI and its application in AS.
- MRI is a valuable tool for clinically significant prostate cancer but can have some applications for lower-grade neoplasms as baseline imaging for AS.
Reviewer 2 Report
Authors should be congratulated for their work. The topic is intriguing and interesting. The manuscript is well-written and methodology is clear. Tables are clear.
I suggest improving the discussion section by adding these 2 important papers (PMID: 33792737; PMID: 33348956)
Author Response
We thank the reviewer for the time spent on the manuscript. We are grateful for the kind words and the favorable review. We are also thankful for the paper's criticism, which has allowed for a much better overall publication. The suggested references have been beneficial reading and aided us in elevating the manuscript.
All the reviewer's suggestions have been addressed as follows:
- English language and medical editing have been carried out by a professional.
- The introduction has been remodified.
- The inclusion of the suggested literature has improved the citations.
- The research design has been enhanced by adding Gleason score values.
- The results are represented more clearly.
- The conclusions section has been improved.
- The suggested essential references have been added and have improved the paper.
Round 2
Reviewer 1 Report
the authors mention - The statistical analysis was performed using SPSS version 27.0 (SPSS Inc., Chicago,
IL, USA). All tests were two-tailed, and the results were interpreted as significant at
Type I error alpha = 0.05 (p < 0.05).
this is insufficient. please mention which all statistical tests were used to do what all comparisons. please refer to any published article in a high impact journal to see how it is conventionally written
Author Response
We want to thank the reviewer for their advice in formulating the statistics description better. We hope that the paper has reached a sufficient standard for publishing.
1. We have added the proper statistical analysis description.
Reviewer 2 Report
Authors should be congratulated for their work. They improved the quality of the manuscript, answering all my concerns. The paper is suitable for publication
Author Response
We want to thank the reviewer again for their kind words.
1. We have added a proper description of the statistical methods used in the research.